# Comorbidities affecting re-admission and survival in COVID-19: Application of joint frailty model

Akram Yazdani[1☯], Seyyed Ali Mozaffarpur[2☯], Pouyan Ebrahimi[3☯], Hoda Shirafkan[4]*, Hamed Mehdinejad[5,6]

1 Department of Biostatistics and Epidemiology, Faculty of Health, Kashan University of Medical Sciences, Kashan, Iran, 2 Traditional Medicine and History of Medical Sciences Research Center, Health Research Institute, Babol University of Medical Sciences, Babol, Iran, 3 Student Research Committee, Babol University of Medical Sciences, Babol, Iran, 4 Social Determinants of Health Research Center, Health Research Institute, Babol University of Medical Sciences, Babol, Iran, 5 Infectious Diseases and Tropical Medicine Research Center, Health Research Institute, Babol University of Medical Sciences, Babol, Iran, 6 Clinical Research Development Unit of Rouhani Hospital, Babol University of Medical Sciences, Babol, Iran

☯ These authors contributed equally to this work.
* hodashirafkan@gmail.com

**Data Availability Statement:** *** PA @ ACCEPT: request non-author contact*** Data are available with the approval of the Ethics Committee of Babol university of medical sciences (contact via

## Abstract

### Background

One of the common concerns of healthcare systems is the potential for re-admission of COVID-19 patients. In addition to adding costs to the healthcare system, re-admissions also endanger patient safety. Recognizing the factors that influence re-admission, can help provide appropriate and optimal health care. The aim of this study was to assess comorbidities that affect re-admission and survival in COVID-19 patients using a joint frailty model.

### Methods

This historical cohort study was done using data of patients with COVID-19 who were re-hospitalized more than twice in a referral hospital in North of Iran. We used the joint frailty model to investigate prognostic factors of survival and recurrence, simultaneously using R version 3.5.1 (library "frailtypack"). P-values less than 0.05 were considered as statistically significant.

### Results

A total of 112 patients with mean (SD) age of 63.76 (14.58) years old were recruited into the study. Forty-eight (42.9%) patients died in which 53.83% of them were re-admitted for a second time. Using adjusted joint model, the hazard of re-admission increased with cancer (Hazard ratio (HR) = 1.92) and hyperlipidemia (HR = 1.22). Furthermore, the hazard of death increased with hyperlipidemia (HR = 4.05) followed by age (HR = 1.76) and cancer (HR = 1.64). It Also decreased with lung disease (HR = 0.11), hypothyroidism (HR = 0.32), and hypertension (HR = 0.97).

hodashirafkan@gmail.com) for researchers who meet the criteria for access to confidential data.

**Funding:** The author(s) received no specific funding for this work.

**Competing interests:** The authors have declared that no competing interests exist.

## Conclusion

Considering the correlation between re-admission and mortality in the joint frailty model, malignancy and hyperlipidemia increased the risk of both re-admission and mortality. Moreover, lung disease probably due to the use of corticosteroids, was a protective factor against both mortality and re-admission.

## Introduction

COVID-19 has caused problems in healthcare systems by causing mortality, morbidities, and anxiety [1]. Many methods of treatment have been introduced to help managing this epidemy [2, 3].

A common concern is the possibility of patients getting re-admitted to hospital. Re-admission, in addition to imposing additional burden on the health care system, also endangers the safety of patients. Recognizing the factors affecting re-admission, management, and policy planning can help provide appropriate and optimal health care [4]. As COVID-19 can have long-lasting implications on the health of individuals, more attention is paid to people who are at a higher risk of developing complications, such as the elderly and people with comorbidities like diabetes mellitus (DM) and hypertension (HTN). Individuals with underlying comorbidities have been found to have poor prognosis [5]. For instance, DM patients were found to have a higher risk of morbidity and mortality, which can lead to hospitalizations and intensive care unit)ICU (admissions [6].

In some diseases, multiple events may occur after initial treatment due to the physical progression of the disease. For example, breast cancer patients may experience breast cancer recurrence, metastasis, or death after initial treatment [7]. In COVID-19 infection, patients may be re-hospitalized after initial treatment and discharge with different causes (including respiratory distress, metabolic encephalopathy, etc.) [4]. Thus, to analyse recurrent events, we propose a shared frailty model taking into account the time to re-admission or death to correlate different events among participants [8].

The conventional frailty model assumes that all patients experience the event of interest with varying non-zero risks [9]. It is an extension of the Cox model to control heterogeneity between individuals [10]. Observations of repeated events in longitudinal studies can be terminated by several types of events, such as study termination, loss to follow-up, or major failure events such as death. In some cases, major failure events may be correlated with the recurrence of major events such as shortness of breath. Therefore, it may violate the assumption of non-informational censorship [11]. This dependency is incorporated into the joint modeling of deaths and recurring events. In this model, the effects of frailty on recurrent events and mortality are not the same, and estimates are made using the Monte Carlo EM algorithm [12].

In this study, we assess the comorbidities that affect re-admission and survival in COVID-19 patients, using penalized probabilities of hazard functions to assess relapse and terminal event processes. We consider the correlation between re-admission and mortality risk through the general gamma-frailty model. Understanding the factors that influence the risk of death in re-admitted patients may help policymakers optimize care.

## Materials and methods

### Study design

This retrospective cohort study was conducted with patients with COVID-19 who were re-admitted after the initial admission and discharge at Ruhani hospital in Babol, Iran from February 2020 to July 2021. Ruhani hospital is the biggest general hospital in North of Iran that covers more than 5 million residents in two Northern provinces of Iran.

All patients with multiple (at least two) hospitalizations for COVID-19 from February 2020 up to July 2021 in Babol, North of Iran were included in the study. All of these patients had follow-ups during their hospital stay. In July 2021, we conducted follow-up calls with patients who had been discharged alive in their last hospitalization to ascertain their current status, particularly regarding mortality.

The recruitment process and collection of study data from the patients' hospital files were conducted by investigators (a pulmonologist and a medical student). Written informed consent was obtained at each admission and verbal informed consent was obtained at the telephone call.

Ethics approval for this study was obtained from the Ethics Committee of the Babol University of Medical Sciences (IR.MUBABOL.REC.1400.187(. The researchers followed the principles of the Helsinki Declaration.

### Participants

Patients who met the following eligibility criteria were recruited into the study.

**Inclusion criteria.**   Patients with age more than 17 years old, confirmation of COVID-19 (positive reverse-transcriptase–polymerase-chain-reaction (RT-PCR)), or having COVID-19 pneumonia (confirmed by imaging), and re-admission due to COVID-19 were recruited for the study.

**Exclusion criteria.**   Patients whose records were not complete, pregnant patients, and patients who were discharged from the hospital with their personal consent were excluded from the study.

**Patient discharge criteria.**   The discharge criteria including no fever for 3 consecutive days, improving in respiratory symptoms, and evident resolution and improvement of the acute lung lesion in imaging was uniform for all patients according to the WHO clinical guidance for COVID-19 [13].

### Study variables

Gender, age, time and duration of hospitalization, mortality, history of underlying comorbidities (including DM, HTN, cancer, heart disease, lung disease, kidney disease, and hyperlipidemia (HLP)), respiratory rate, oxygen saturation, and having any supplemental oxygen therapy were recorded from hospital records. Assessing the severity of the disease, we used national early warning score for COVID-19 (NEWS2) [14]. NEWS2 although have some limitations in moderate severity of COVID-19, is the best available scoring method that has 8 questions, and each question is scored between 0 to 3 (0 indicate normal values and 3 for the worst values). The total score of NEWS2 is categorized as low for scores 0 to 4, medium for scores 5 to 6 and more than 7 as high and score of 3 in any individual parameter as low-medium. Evaluating the current status of mortality, we followed all patients after the last discharge by a telephone call in June and July 2021.

## Statistical analysis

Descriptive statistics were presented as mean (standard deviation-SD), mean difference (MD), median, mode, interquartile range (IQR), and frequency (percent) for quantitative and qualitative data, respectively. We assessed the normality of variables by the Kolmogorov-Smirnov test. To evaluate the characteristics and comorbidities between survivors and dead patients we used independent t-test and chi-square test. We used the joint frailty model to investigate prognostic factors of survival and recurrence, simultaneously.

**Joint frailty model.**   The joint frailty model models the death and recurrent event processes simultaneously by a shared random effect. In summary, the hazard functions for the recurrent event and the death processes at time t can be expressed as

$$
\begin{aligned}
r_{ij}(t \mid v_i) &= r_0(t)\exp(\beta \boldsymbol{Z}_i + v_i) \\
\lambda_i(t \mid v_i) &= \lambda_0(t)\exp(\alpha \boldsymbol{Z}_i + g v_i)
\end{aligned}
\tag{1}
$$

in which i shows patient and j the recurrent event, we show the vector of covariate for each patient as $Z_i$, and $\alpha$ and $\beta$ as the regression coefficient for the recurrent event and death, respectively. In this model, g estimates the correlation between death and recurrent event processes. The baseline (underlying) hazard function for recurrent event and death are represented as $r_0(t)$ and $\lambda_0(t)$, respectively. Furthermore, we show the shared random effect as $v_i$. We assume the distribution of $v_i$ is normal with mean 0 and variance of j. This variance shows the unobservable heterogeneity between patients.

The statistical analysis was done using R version 3.5.1 (function " frailtyPenal" in the library "frailtypack"). P-values less than 0.05 were considered as statistically significant.

## Results

A total of 112 patients with the mean (SD) age of 63.76 (14.58) (median: 65 IQR:51 to 72; 75 (67%) more than 60 years old) were recruited into the study; from which 48 (42.9%) patients died. Fifty-three and eighty-three percent (53.83%) of mortalities were at the second re-admission with an average of 76 days (SD: 108, median: 30, IQR: 7 to 273, mode = 13 days) after the first hospitalization. Also, 53 (47.32%) patients were re-hospitalized at least 3 times and 30 patients received ICU care at least once. Sixty-nine (61.6%) patients were male. The median length of hospital stay was 6 days (IQR: 7: 3 to 10, mean (SD): 8.33(8.63)). Patients have been hospitalized with a total of 363 patient-time and an average (SD) of 2.70 (2.01) times (Table 1).

**Table 1. Frequency of re-admissions, survival and death among patients.**

|  | Re-admission | Survived | Dead |
|---|---|---|---|
|  | (n = 112) | (n = 64) | (n = 48) |
| First hospitalization | - | 112 | 0 |
| Second hospitalization | 112 | 89 (79.5%)* | 23 (20.5%) |
| Third hospitalization | 53 | 46 (86.8%) | 7 (13.2%) |
| Fourth hospitalization | 31 | 24 (77.4%) | 7 (22.6%) |
| Fifth hospitalization | 18 | 16 (78.6%) | 2 (21.4%) |
| Sixth hospitalization | 14 | 11 (78.6%) | 3 (21.4%) |
| Seventh hospitalization | 10 | 8 (80.0%) | 2 (20.0%) |
| More than eight hospitalizations | 4 | 0 | 4 (100%) |

*The percentages are within each admission (row percentages)

Comparing the baseline characteristics of the patients, no differences were observed between age and gender of patients between groups (group A: patients who were discharged and group B: patients who died). Assessing underling diseases between two groups, we saw that the mortality rate among patients with cancer was higher than patients without cancer (OR = 2.79, p-value = 0.04), (Table 2).

Assessing the severity of COVID-19 by NEWS2, we found that majority of patients were at high severity (n = 39, 36.8%) in the first hospitalization (the severity of COVID-19 at low, low to medium, and medium were 24 (22.6%), 18 (17.0%), and 25 (23.6%) patients, respectively). Out of 39 patients with high severity, 53.8% patients died.

## Primary symptoms

The most common COVID-19 symptoms among 363 patient-times were shortness of breath in 154 (42.42%), fever in 92 (25.34%), cough in 77 (21.21%) and fatigue in 108 (29.75%).The other symptoms were sore throat (n = 2, 0.5%), chills (n = 72,19.83%), headache (n = 24, 6.61%), diarrhea (n = 9, 2.47%), chest pain (n = 21, 5.78%), sweating (n = 11, 3.30%), abdominal pain (n = 27, 7.44%), myalgia (n = 23, 6.34%), loss of appetite (n = 51, 14.05%), nausea and vomiting (n = 56, 15.43%), and impaired consciousness (n = 22, 6.06%).

The number of re-admissions varied from 1 to 13, with the average of 2.70 and median of 2 times per patient. We categorized episodes into the number of observation times for each patient. The last episode corresponds to a death or censoring time (discharge or after the last follow-up through the phone call).

## Prognostic factors for re-admission and mortality

Table 3 presents the results using adjusted joint model. We run several models. In the first model we inter only gender and age (primary model). In the other models, we added one of the underlying disease parameters (HLP, kidney disease, etc.) to the primary model. The hazard of recurrence of re-admission increased with cancer (HR: 1.92, confidence interval 95% (95%CI): (1.61,2.23)) and with HLP (HR:1.22, 95%CI: (0.79, 1.65)).

**Table 2. Baseline characteristics and their comparisons between surviving status of COVID-19 patients with re-admissions.**

|  | Total* | Died | Discharged | p-value |
|---|---|---|---|---|
|  | (n = 112) | (n = 48) | (n = 64) |  |
| Sex (Male) | 69 (61.6%) | 29 (60.4%) | 43 (62.5%) | 0.822 |
| Age (year) Mean(SD) | 63.77(14.58) | 65.48(14.16) | 62.58(14.72) | 0.297 |
| HLP[a] | 19 (17.00%) | 11 (23.4%) | 8 (12.5%) | 0.132 |
| Kidney diseases | 13 (11.6%) | 8 (17.0%) | 5 (7.8%) | 0.135 |
| Lung diseases | 13 (11.6%) | 3 (4.7%) | 10 (15.7%) | 0.135 |
| Heart diseases | 48 (42.9%) | 21 (44.7%) | 27 (42.2%) | 0.793 |
| Cancer | 19 (17.0%) | 12 (25.5%) | 7 (10 .9%) | **0.044** |
| Hypothyroidism | 9 (8.0%) | 3 (6.4%) | 6 (9.4%) | 0.568 |
| DM[b] | 52 (46.4%) | 24 (50.0%) | 28 (43.8%) | 0.445 |
| HTN[c] | 49 (43.8%) | 21 (44.7%) | 28 (43.8%) | 0.922 |

* All percentages are column percent and significant p-values are shown as bold text.

[a]HLP: Hyperlipidemia

[b]DM: Diabetes mellitus

[c]HTN: Hypertension

**Table 3. Factors affecting recurrent of hospitalization and survival of COVID-19 patients using joint frailty model.**

| Variables | | Recurrent event (Re-hospitalization) | | Terminal event (Death) | |
|---|---|---|---|---|---|
| | | HR (95%CI) | p-value | HR (95%CI) | p-value |
| Primary model | Gender (male) | 0.79 (0.496,1.08) | 0.144 | 0.67 (0.33,1.36) | 0.290 |
| | Age | 0.90 (0.76,1.05) | 0.207 | 1.76 (1.09,2.82) | 0.022 |
| HLP[a] | | 1.22 (0.79,1.87) | 0.363 | 4.05 (1.11,14.76) | 0.035 |
| Kidney diseases[a] | | 0.81 (0.51,1.29) | 0.397 | 1.74 (0.52,5.86) | 0.369 |
| Lung disease[a] | | 0.61 (0.36,1.03) | 0.074 | 0.11 (0.02,0.67) | 0.019 |
| Heart disease[a] | | 0.92 (0.68,1.23) | 0.603 | 0.88 (0.42,1.85) | 0.754 |
| Cancer[a] | | 1.92 (1.40,2.63) | <0.001 | 1.64 (0.71,3.81) | 0.254 |
| Hypothyroidism[a] | | 0.96 (0.57,1.60) | 0.896 | 0.32 (0.06,1.60) | 0.181 |
| DM[a] | | 0.89 (0.65,1.22) | 0.499 | 4.77 (1.83,12.46) | 0.001 |
| HTN[a] | | 0.98 (0.73,1.31) | 0.933 | 0.97 (0.43,2.17) | 0.943 |

a: Adjusted for the age and gender.

The hazard of death increased with cancer (HR = 1.64, 95%CI (0.71,3.81)), DM (HR = 4.77, 95%CI (1.83,12.46)), and HLP (HR = 4.05, 95%CI (1.11,14.76)). Also, it decreased with lung disease (HR = 0.11, 95%CI (0.02,0.67)), hypothyroidism (HR = 0.32, 95%CI (0.06,1.60)), and HTN (HR = 0.97, 95%CI (0.43,2.17)). Furthermore, the hazard of death increased with increasing age (HR = 1.76, 95%CI (1.09,2.82)). The baseline survival functions for death and recurrent events (joint frailty model) were shown in Fig 1. The data reveals a clear pattern: as the number of re-admissions increases, survival rates exhibit a more pronounced decline over time.

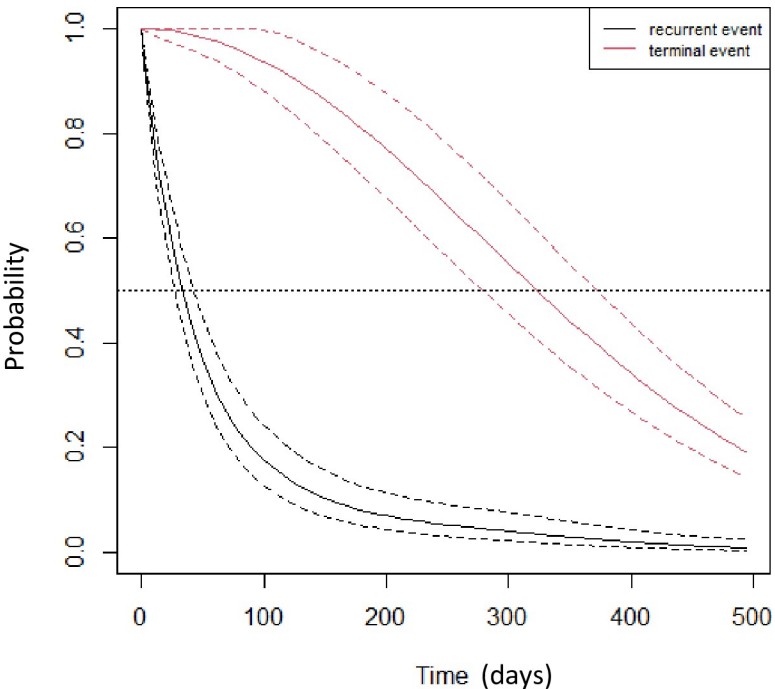

**Fig 1. Baseline survival functions for mortality and recurrent events (joint frailty model).**

## Discussion

In this study, we used a joint frailty model to find prognostic factors for the survival of COVID-19 patients considering the correlation with the recurrence of their re-hospitalization. Based on our best knowledge, the joint study of re-admission and survival among COVID-19 patients has not been done, yet. Using this model, we found that malignancy and hyperlipidemia were risk factors for both the occurrence of mortality and also re-admission. Moreover, chronic lung disease was a protective factor against mortality and recurrent hospitalization. We found 112 confirmed cases, which needed re-admission. We did not see any differences in age and gender between patients who were discharged and died. In a meta-analysis, Clara Bonanad et al (2020) emphasized the effect of age on mortality with the thresholds of age >50 years and, particularly, >60 years [15]. The effect of age on mortality has also been shown in other studies [16], too. In our study, the hazard of mortality increased with age.

In our study cancer increased the hazard of both recurrent admissions and mortality. The hazard of death and re-hospitalization in cancer patients was respectively 64% and 92% higher than that of patients without a history of cancer. It was seen that among readmitted patients, malignancy history was more prevalent. Especially, those patients who were readmitted to thrombotic episodes had a history of Malignancy, emphasizing the contribution of malignancy to the already existing hypercoagulability risk in COVID-19 [2]. L. Zhang et al (2020) observed that cancer patients display deteriorating situations and negative consequences from the COVID-19 infection [17]. Ludi Yang et al (2020) in a systematic review study examined the consequences of cancer in COVID-19 patients. They showed that cancer patients were more susceptible to COVID-19. As a risk factor, cancer increased mortality in COVID-19 patients. Among cancer patients with COVID-19, those with lung cancer had a higher mortality rate than those without lung cancer [18].

Chronic lung disease was more common in patients who required re-admission than in those who did not, highlighting its potential impact on the development of dyspnea. It is a common cause of COPD exacerbations and increases the risk of subsequent re-admission [19]. Jaime Signes-Costa et al. (2021) showed that previous lung disease was a risk factor for mortality in patients with COVID-19 [20]. Jacobo Rogado et al. (2020) showed that COVID-19 patients who already had lung cancer have a higher mortality rate than the general population [21]. In our study lung disease did not have significant effect on re-hospitalization but the effect size was considerable. Furthermore, it decreased the hazard of mortality which could be due to the use of corticosteroid due to the lung disease before contracting COVID-19 [22].

In our study hypothyroidism disease and high hypertension decreased the hazard of mortality and did not have any significant assertion with re-hospitalization. So-Young Kim examined the association between susceptibility, morbidity and mortality due to COVID-19 and hypothyroidism disease and found that a history of hypothyroidism disease was not associated with an increased risk of morbidity and mortality of COVID-19 [23]. Furthermore, Yanbin Du et al. (2020) in their meta-analysis proposed that hypertension was independently associated with a significantly increased risk of severe COVID-19 and also in-hospital COVID-19 [24].

Furthermore, we found that hyperlipedemia increased the risk of mortality. It is concordant with other studies [25–27]. Its possible mechanism is that when the lung is involved and the blood oxygen level decreases, one of the main compensatory mechanisms of the body to deliver enough oxygen to the tissues is to increase the cardiac output by increasing the heart rates and increasing the stroke volume. Naturally, endothelial dysfunction, atherosclerosis, and possible narrowing of coronary arteries in patients with hyperlipidemia, as well as probable more coronary artery diseases in them, can reduce the ability of the cardiovascular system to compensate tissue hypoxia in these patients. Also, three mechanisms of increased oxidative

stress, pro-inflammatory conditions, and also risk of arrhythmia caused by hyperlipidemia [28] combined with COVID-19 can lead to increased mortality.

### Advantages of the study

An advantage of our study is the ability to analyze recurring events and associated terminating event data simultaneously by joint frailty model. Another advantage of our study is assessing effect of the history of underlying comorbidities (including DM, HTN, cancer, heart disease, lung disease, kidney disease, and hyperlipidemia (HLP)) for re-admission and survival.

### Limitations of the study

We entered into the study all patients that were suspected of COVID-19 or whose RT-PCR was positive. As a limitation of our study, we did not assess the lung and chest imaging findings of the re-admitted patients, this was beyond the scope of our research. Another limitation of our study, we consider all re-admitted patients at a hospital in Babol in the north of Iran. This is a referral COVID-19 hospital, and patients with worse conditions refer to this hospital. This might affect the generalizability of the study results. As a suggestion for more studies, re-admitting patients with narrower eligibility criteria could have more consistent results.

As another limitation, some patients could have fallen out of the study observation without being admitted to our hospital system since the follow-up ended with the end of the study. But our claim is not more that the study time.

### Conclusion

In our cohort study, about half of the patients were re-admitted at least 3 times and the first re-admission for them was within the first month after the first discharge. This is probably tended to be associated with the index medical institution course and eventually to COVID-19, highlighting that some re-admissions were due to latent errors or omissions in hospital care. Considering the correlation with the re-hospitalization and mortality in a joint frailty model, malignancy and HLP increased the hazard of both re-hospitalization and mortality. Furthermore, having lung diseases was a protective factor for both mortality and re-hospitalization. Therefore, advanced statistical models can be used to more accurately identify risk factors for re-admission and mortality, which can be used in continuing care to prevent patient re-admission and death.

### Acknowledgments

The authors of this article express their appreciation and gratitude to the student research committee of Babol university of medical sciences and the clinical research development center of Rouhani hospital.

### Author Contributions

**Conceptualization:** Akram Yazdani, Seyyed Ali Mozaffarpur, Hoda Shirafkan, Hamed Mehdinejad.

**Data curation:** Pouyan Ebrahimi, Hoda Shirafkan.

**Formal analysis:** Akram Yazdani, Hoda Shirafkan.

**Investigation:** Seyyed Ali Mozaffarpur, Hoda Shirafkan, Hamed Mehdinejad.

**Methodology:** Hoda Shirafkan.

**Project administration:** Hoda Shirafkan.

**Software:** Hoda Shirafkan.

**Supervision:** Seyyed Ali Mozaffarpur, Hamed Mehdinejad.

**Validation:** Hoda Shirafkan.

**Writing – original draft:** Seyyed Ali Mozaffarpur, Hoda Shirafkan.

**Writing – review & editing:** Akram Yazdani, Seyyed Ali Mozaffarpur, Pouyan Ebrahimi, Hamed Mehdinejad.

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
