## [Decision Letter · Decision Letter 0]

24 Apr 2023

PONE-D-23-04746Comorbidities affecting re-admission and survival in COVID-19: Application of joint frailty modelPLOS ONE

Dear Dr. Shirafkan,

Thank you for submitting your manuscript to PLOS ONE. After careful consideration, we feel that it has merit but does not fully meet PLOS ONE’s publication criteria as it currently stands. Therefore, we invite you to submit a revised version of the manuscript that addresses the points raised during the review process.

This is an observational study and the authors may want to consider using the STROBE guidelines to report this 

https://www.equator-network.org/reporting-guidelines/strobe/==============================

We look forward to receiving your revised manuscript.

Kind regards,

Yee Gary Ang, MBBS MPH

Academic Editor

PLOS ONE

Journal Requirements:

Reviewers' comments:

Reviewer's Responses to Questions

**Comments to the Author**

1. Is the manuscript technically sound, and do the data support the conclusions?

Reviewer #1: Yes

Reviewer #2: Partly

2. Has the statistical analysis been performed appropriately and rigorously? 

Reviewer #1: No

Reviewer #2: Yes

3. Have the authors made all data underlying the findings in their manuscript fully available?

Reviewer #1: No

Reviewer #2: Yes

4. Is the manuscript presented in an intelligible fashion and written in standard English?

Reviewer #1: Yes

Reviewer #2: No

5. Review Comments to the Author

Reviewer #1: Dear Author/s

Greetings

I appreciate your effort and wish you healthy and happy days.

There are some shortcomings in the article.

-You said '' The hazard of death increased with cancer (HR=1.64, SE=0.43) and HLP 197

(HR=4.05, SE=0.66) and decreased with lung disease (HR=0.11, SE=0.92), 198

hypothyroidism (HR=0.32, SE= 0.82) and HTN (HR=0.97, SE=0.41). Also,the hazard 199

of death increased with increase age (HR=1.76, SE=0.24).'' but Table 3 shows that HLP, age, and DM increase mortality, while lung disease decreases it.

-It is not very significant that lung disease reduces mortality.

The article is unfortunately not suitable for publication.

Best regards

Reviewer #2: Firstly, I thank the editor for the opportunity to review this paper. I also commend the authors for addressing the important issue of readmissions among COVID-19 patients, which has huge implications on the capacity of health systems to cope. It was a decent attempt and the findings were informative. However, I had some difficulties with the way the paper was written and the quality of the language used. There are grammatical and sentence structure issues. More attention needs to be paid to the use of scientific language in this paper. I also had concerns with the methodology, particularly, the recruitment of patients into the study. The details of how the study was done, at what frequency were the patients contacted, how was informed consent obtained and how patient data was managed and the steps taken to ensure data privacy was not specified clearly at all. I have stated my detailed comments in the attached file. Overall, the paper needs more work.

6. PLOS authors have the option to publish the peer review history of their article (what does this mean?). If published, this will include your full peer review and any attached files.

Reviewer #1: No

Reviewer #2: No

---

## [Author Response · Author response to Decision Letter 0]

15 Jun 2023

Reviewer #1:

comment1:

You said '' The hazard of death increased with cancer (HR=1.64, SE=0.43) and HLP (HR=4.05, SE=0.66) and decreased with lung disease (HR=0.11, SE=0.92), hypothyroidism (HR=0.32, SE= 0.82) and HTN (HR=0.97, SE=0.41). Also, the hazard of death increased with increase age (HR=1.76, SE=0.24).'' but Table 3 shows that HLP, age, and DM increase mortality, while lung disease decreases it.

Response: we change the manuscript text as follow:

Table 3 presents the results using adjusted joint model. The hazard of recurrence of readmission increased with cancer (HR=1.92, Standard error [SE]=0.16) and with HLP (HR=1.22, SE=0.22).

And Also:

The hazard of death increased with cancer (HR=1.64, SE=0.43), DM (HR=4.77, SE=0.49), and HLP (HR=4.05, SE=0.66). Also, it decreased with lung disease (HR=0.11, SE=0.92), hypothyroidism (HR=0.32, SE= 0.82), and HTN (HR=0.97, SE=0.41). Furthermore, the hazard of death increased with increasing age (HR=1.76, SE=0.24).

Comment2:

It is not very significant that lung disease reduces mortality.

Response: Considering that the p-value depends on the sample size, it is better to pay attention to the effect size in the interpretation of the results. In the results of the present study, although the p-value was not significant, but considering that the effect size was large, we used the word considerable in its interpretation.

“In our study lung disease did not have significant effect on re-hospitalization but the effect size was considerable.” (Line 250)

Reviewer #2: 

Comment1: 

However, I had some difficulties with the way the paper was written and the quality of the language used. There are grammatical and sentence structure issues. More attention needs to be paid to the use of scientific language in this paper.

Response: We re-check the manuscript text. English editing by a native English was done.

Comment2:

I also had concerns with the methodology, particularly, the recruitment of patients into the study. The details of how the study was done, at what frequency were the patients contacted, how was informed consent obtained and how patient data was managed and the steps taken to ensure data privacy was not specified clearly at all. I have stated my detailed comments in the attached file. Overall, the paper needs more work

Response: In response to “how were the patients recruited?” we added the following to the manuscript:

All patients with multiple (at least two) hospitalizations for Covid-19 since the beginning of the Covid pandemic in Iran (February 2020) were included in the study. All of these patients had follow-ups during their hospital stay and until the end of the study (July 2021). For some of these patients, he was hospitalized more than once during the study period and that information was also collected. In July 2021, we called the patient who was discharged alive during his last hospitalization and obtained the patient's last status (mortality) on this call. 

In response to “recruited by who, were the recruiters trained researchers, were participants provided with a participant information sheet, was consent verbal or written etc.?”, we added the following to the manuscript.

The recruitment process and collection of study data were conducted by investigators (A pulmonologist and A medical student). Written informed consent was obtained at each admission and verbal informed consent was obtained at the telephone call.

---

## [Decision Letter · Decision Letter 1]

7 Jul 2023

PONE-D-23-04746R1Comorbidities affecting re-admission and survival in COVID-19: Application of joint frailty modelPLOS ONE

Dear Dr. Shirafkan,

Thank you for submitting your manuscript to PLOS ONE. After careful consideration, we feel that it has merit but does not fully meet PLOS ONE’s publication criteria as it currently stands. Therefore, we invite you to submit a revised version of the manuscript that addresses the points raised during the review process.

Please kindly read through the reviews provided and address the concerns as one of the reviewers mentioned that the concerns had not been addressed adequately

We look forward to receiving your revised manuscript.

Kind regards,

Yee Gary Ang, MBBS MPH

Academic Editor

PLOS ONE

Reviewers' comments:

Reviewer's Responses to Questions

**Comments to the Author**

1. If the authors have adequately addressed your comments raised in a previous round of review and you feel that this manuscript is now acceptable for publication, you may indicate that here to bypass the “Comments to the Author” section, enter your conflict of interest statement in the “Confidential to Editor” section, and submit your "Accept" recommendation.

Reviewer #2: (No Response)

Reviewer #3: (No Response)

2. Is the manuscript technically sound, and do the data support the conclusions?

Reviewer #2: Partly

Reviewer #3: Partly

3. Has the statistical analysis been performed appropriately and rigorously? 

Reviewer #2: Yes

Reviewer #3: I Don't Know

4. Have the authors made all data underlying the findings in their manuscript fully available?

Reviewer #2: Yes

Reviewer #3: Yes

5. Is the manuscript presented in an intelligible fashion and written in standard English?

Reviewer #2: No

Reviewer #3: (No Response)

6. Review Comments to the Author

Reviewer #2: The comments from the previous rounds of revisions have not been addressed comprehensively. There are still fundamental flaws with the methodology with insufficient information provided on how the participants were recruited (were they automatically recruited when they presented at the hospital for the second admission? When did recruitment take please? Were participants provided with a information sheet and consent form and the study explained to them before they signed the written consent form?). Moreover, there are still issues with language and grammatical errors that should be fixed for clarity. The methodology section more work, particularly, in describing what the NEWS2 socres indicate. In addition, the findings are not thoroughly discussed in the discussion section and citations need to be included to support claims made in the discussion. Overall, the manuscript still needs considerable revision before it is of publishable quality. Please refer to the attached manuscript file for detailed comments and suggested edits. Thank you.

Reviewer #3: Comments to the authors:

I commend the authors for exploring the important topic of readmission after the COVID-19 disease. The authors performed a comprehensive collection of the available data. Using the joint frailty model is a strength of this study.

Abstract:

The abstract needs to be structured more fluently and use consistent terminology (e.g., “re-admission” versus “readmission”). Some numbers need to be spelled out (more than twice versus “ more than 2 times”). I suggest introducing the hazard of death influence in the order of strength: hyperlipidemia (HR 4.05), followed by age (HR 1.76), and cancer (1.64).

Introduction:

This part is improved following the suggestions and comments from “A.” However, some inconsistencies regarding terminology and grammar errors exist.

Material and Methods:

Study design:

The inclusion criteria for this study are marked by bias. Some patients could have fallen out of the study observation without being admitted to the same hospital system since the follow-up ended with the end of the study (July 2021).

Study variables:

Using the NEWS2 score to assess the severity of COVID-19 can be debatable since the authors of the article that introduced the score mentioned poor-to-moderate discrimination for medium-term COVID-19 outcomes.

Results:

The data is presented in a complicated, overfilled-with-numbers manner that is difficult to follow. Again, I suggest consistency in terminology; there are too many terms of the same meaning ("re-admission," "readmission," "re-hospitalization").

Discussion:

The authors should give the most plausible explanation as to why the history of lung diseases was protective. I also recommend being consistent and using either lower or capital letters for diseases such as malignancy.

7. PLOS authors have the option to publish the peer review history of their article (what does this mean?). If published, this will include your full peer review and any attached files.

Reviewer #2: No

Reviewer #3: No

---

## [Author Response · Author response to Decision Letter 1]

25 Aug 2023

Reviewer #2: 

 Please refer to the attached manuscript file for detailed comments and suggested edits

Response to reviewer #2:

We tried to response and correct all the comments. All the corrections are done in the attached manuscript.

Reviewer #3: 

Title: Comorbidities affecting re-admission and survival in COVID-19: Application of joint frailty model

Comments to the authors: 

I commend the authors for exploring the important topic of readmission after the COVID-19 disease. The authors performed a comprehensive collection of the available data. Using the joint frailty model is a strength of this study. 

Abstract: 

The abstract needs to be structured more fluently and use consistent terminology (e.g., “re-admission” versus “readmission”). Some numbers need to be spelled out (more than twice versus “ more than 2 times”). I suggest introducing the hazard of death influence in the order of strength: hyperlipidemia (HR 4.05), followed by age (HR 1.76), and cancer (1.64).

Response: we corrected them

Introduction: 

This part is improved following the suggestions and comments from “A.” However, some inconsistencies regarding terminology and grammar errors exist. 

Response: we tried to corrected them

Material and Methods: 

Study design: 

The inclusion criteria for this study are marked by bias. Some patients could have fallen out of the study observation without being admitted to the same hospital system since the follow-up ended with the end of the study (July 2021). 

Response: we add this point in the limitations of the study

Study variables: 

Using the NEWS2 score to assess the severity of COVID-19 can be debatable since the authors of the article that introduced the score mentioned poor-to-moderate discrimination for medium-term COVID-19 outcomes.

Response: we mention this point in the method

Results: 

The data is presented in a complicated, overfilled-with-numbers manner that is difficult to follow. Again, I suggest consistency in terminology; there are too many terms of the same meaning ("re-admission," "readmission," "re-hospitalization"). 

Response: we tried to corrected them

Discussion:

The authors should give the most plausible explanation as to why the history of lung diseases was protective. I also recommend being consistent and using either lower or capital letters for diseases such as malignancy.

Response: we corrected them. Also, the probable physiopathology of our finding was declared (use of corticosteroids)

---

## [Decision Letter · Decision Letter 2]

10 Nov 2023

PONE-D-23-04746R2Comorbidities affecting re-admission and survival in COVID-19: Application of joint frailty modelPLOS ONE

Dear Dr. Shirafkan,

Thank you for submitting your manuscript to PLOS ONE. After careful consideration, we feel that it has merit but does not fully meet PLOS ONE’s publication criteria as it currently stands. Therefore, we invite you to submit a revised version of the manuscript that addresses the points raised during the review process.

We have sent your manuscript to 3 reviewers, 1 of which has rejected it

However, we invite you to revise extensively the manuscript before submission  

We look forward to receiving your revised manuscript.

Kind regards,

Yee Gary Ang, MBBS MPH

Academic Editor

PLOS ONE

Reviewers' comments:

Reviewer's Responses to Questions

**Comments to the Author**

1. If the authors have adequately addressed your comments raised in a previous round of review and you feel that this manuscript is now acceptable for publication, you may indicate that here to bypass the “Comments to the Author” section, enter your conflict of interest statement in the “Confidential to Editor” section, and submit your "Accept" recommendation.

Reviewer #3: (No Response)

Reviewer #4: All comments have been addressed

Reviewer #5: (No Response)

2. Is the manuscript technically sound, and do the data support the conclusions?

Reviewer #3: Partly

Reviewer #4: Yes

Reviewer #5: No

3. Has the statistical analysis been performed appropriately and rigorously? 

Reviewer #3: I Don't Know

Reviewer #4: Yes

Reviewer #5: No

4. Have the authors made all data underlying the findings in their manuscript fully available?

Reviewer #3: No

Reviewer #4: No

Reviewer #5: Yes

5. Is the manuscript presented in an intelligible fashion and written in standard English?

Reviewer #3: No

Reviewer #4: No

Reviewer #5: No

6. Review Comments to the Author

Reviewer #3: Title: Comorbidities affecting re-admission and survival in COVID-19: Application of joint frailty model

Comments to the authors:

Again, I commend the authors for exploring the important topic of readmission after the COVID-19 disease. The authors performed a comprehensive collection of the available data. Using the joint frailty model is a strength of this study. The authors addressed many of the reviewers’ suggestions. Unfortunately, the clarity and fluidity in English continue to be a problem overall, and I recommend using an editor who can proofread and correct the English semantics and grammar.

Abstract:

The authors incorporated the suggestions for the abstract. The content reads more clearly and concisely.

Introduction:

Please correct linguistic inconsistencies and grammar errors. Consider using a native English speaker editor.

For example the line 58-59 in the Introduction should read “ Recognizing the factors affecting re-admission, management, and policy planning can help provide appropriate and optimal health care” and 70-72 should read “In COVID-19 infection, patients may be re-hospitalized after initial treatment and discharge with different causes (including respiratory distress, metabolic encephalopathy, etc.)”

Material and Methods:

Study design:

Some patients could have fallen out of the study observation without being admitted to the same hospital system since the follow-up ended with the end of the study (July 2021). The authors addressed this in the discussion as a study limitation.

Study variables:

Using the NEWS2 score to assess the severity of COVID-19 can be debatable since the authors of the article that introduced the score mentioned poor-to-moderate discrimination for medium-term COVID-19 outcomes. This was addressed in one sentence in the Methods (line 132-133).

Results:

The data continues to be presented in a complicated, overfilled-with-numbers manner that is difficult to follow. I suggest using the same pattern they used in the abstract: for instance in the Abstract the authors used “mean (SD) age of 63.76 (14.58)” versus” mean (±SD) age of 63.76 (±14.58)” in the results.

Discussion:

The Discussion part should be more elaborated, not only with insertions from medical literature but to address possible explanations for their study results. There are significant linguistic inaccuracies that need to be corrected as well.

Reviewer #4: Thank you for the opportunity to review this manuscript. Despite with limited sample size and a relatively selected sample, I think this is a nice study that shows how the joint frailty model can be applied in clinical data to explore two outcomes simultaneously. The authors have also responded to the previous reviewers' comments satisfactorily.

Some minor points to be considered and fixed before publication:

- It is not clear how the study variables are considered in the models. For example, are oxygen saturation and NEWS2 included as covariates in the model? Please specify clearly what variables are included in the model in the section "Joint frailty model".

- If possible, I would suggest presenting 95% confidence intervals rather than SE, which is more informative.

- In Figure 1, please specify the unit of time (is it in terms of days?). The label for y-axis should also be changed, as it should be a probablity instead of "baseline survival function"

- There are still some errors all over the places, e.g., line 54: "morbiditiesv,"; line 134: "queation"; line 204: "Prognostig". I suggest the authors proof-read the article again very carefully.

Reviewer #5: The article is written in a very poor and non-standard way

The main article is based on the fitting of the frailty model, but the estimation result of this coefficient has not been reported at all

The confidence intervals of the coefficients are very important, but they are not reported in the article

In one part of the article, you reported the software was used is R and in another section you wrote the STATA software

7. PLOS authors have the option to publish the peer review history of their article (what does this mean?). If published, this will include your full peer review and any attached files.

Reviewer #3: No

Reviewer #4: No

Reviewer #5: No

---

## [Author Response · Author response to Decision Letter 2]

6 Dec 2023

Reviewer #3: 

Comments to the authors:

Again, I commend the authors for exploring the important topic of readmission after the COVID-19 disease. The authors performed a comprehensive collection of the available data. Using the joint frailty model is a strength of this study. The authors addressed many of the reviewers’ suggestions. Unfortunately, the clarity and fluidity in English continue to be a problem overall, and I recommend using an editor who can proofread and correct the English semantics and grammar.

Abstract:

The authors incorporated the suggestions for the abstract. The content reads more clearly and concisely.

Introduction:

Please correct linguistic inconsistencies and grammar errors. Consider using a native English speaker editor.

For example the line 58-59 in the Introduction should read “Recognizing the factors affecting re-admission, management, and policy planning can help provide appropriate and optimal health care” and 70-72 should read “In COVID-19 infection, patients may be re-hospitalized after initial treatment and discharge with different causes (including respiratory distress, metabolic encephalopathy, etc.)”

Response: we tried to correct them. We carefully read and proof the entire text and tried to correct all of the items. In addition, the manuscript was read and corrected by an English native speaker.

Material and Methods:

Study design:

Some patients could have fallen out of the study observation without being admitted to the same hospital system since the follow-up ended with the end of the study (July 2021). The authors addressed this in the discussion as a study limitation.

Study variables:

Using the NEWS2 score to assess the severity of COVID-19 can be debatable since the authors of the article that introduced the score mentioned poor-to-moderate discrimination for medium-term COVID-19 outcomes. This was addressed in one sentence in the Methods (line 132-133).

Results:

The data continues to be presented in a complicated, overfilled-with-numbers manner that is difficult to follow. I suggest using the same pattern they used in the abstract: for instance, in the Abstract the authors used “mean (SD) age of 63.76 (14.58)” versus” mean (±SD) age of 63.76 (±14.58)” in the results.

Response: we tried to correct them. We changed all of the “mean (±SD)”s to the “mean (SD)”.

Discussion:

The Discussion part should be more elaborated, not only with insertions from medical literature but to address possible explanations for their study results. There are significant linguistic inaccuracies that need to be corrected as well.

Response: We tried to add some more explanations about the obtained results. We highlighted the changes in the text.

Reviewer #4: 

Thank you for the opportunity to review this manuscript. Despite with limited sample size and a relatively selected sample, I think this is a nice study that shows how the joint frailty model can be applied in clinical data to explore two outcomes simultaneously. The authors have also responded to the previous reviewers' comments satisfactorily.

Some minor points to be considered and fixed before publication:

- It is not clear how the study variables are considered in the models. For example, are oxygen saturation and NEWS2 included as covariates in the model? Please specify clearly what variables are included in the model in the section "Joint frailty model".

Response: Thanks for the reviewer's comment. We tried to add explanations about how to make models to the text. We added the following statement.

“We run several models. In the first model we inter only gender and age (primary model). In the other models, we added one of the underlying disease parameters (HLP, kidney disease, etc.) to the primary model.”

- If possible, I would suggest presenting 95% confidence intervals rather than SE, which is more informative.

Response: we tried to correct them. We changed all of the “SE”s to the “95%CI”.

- In Figure 1, please specify the unit of time (is it in terms of days?). The label for y-axis should also be changed, as it should be a probability instead of "baseline survival function"

Response: we tried to correct it.

- There are still some errors all over the places, e.g., line 54: "morbiditiesv,"; line 134: "queation"; line 204: "Prognostig". I suggest the authors proof-read the article again very carefully.

Response: we tried to correct them. We carefully read the entire text and tried to correct all the items. In addition, the manuscript was read and corrected by an English native speaker.

Reviewer #5:

 The article is written in a very poor and non-standard way

-The main article is based on the fitting of the frailty model, but the estimation result of this coefficient has not been reported at all.

Response: we fitted a joint frailty model to the data and report all of the coefficients of the models in the table 3

-The confidence intervals of the coefficients are very important, but they are not reported in the article.

Response: we tried to correct them. We changed all of the “SE”s to the “95%CI”.

-In one part of the article, you reported the software was used is R and in another section you wrote the STATA software.

Response: We corrected it.

---

## [Decision Letter · Decision Letter 3]

18 Dec 2023

PONE-D-23-04746R3Comorbidities affecting re-admission and survival in COVID-19: Application of joint frailty modelPLOS ONE

Dear Dr. Shirafkan,

Thank you for submitting your manuscript to PLOS ONE. After careful consideration, we feel that it has merit but does not fully meet PLOS ONE’s publication criteria as it currently stands. Therefore, we invite you to submit a revised version of the manuscript that addresses the points raised during the review process.

We have invited 3 reviewers and we invite you to address their comments and resubmit.

We look forward to receiving your revised manuscript.

Kind regards,

Yee Gary Ang, MBBS MPH

Academic Editor

PLOS ONE

Journal Requirements:

Reviewers' comments:

Reviewer's Responses to Questions

**Comments to the Author**

1. If the authors have adequately addressed your comments raised in a previous round of review and you feel that this manuscript is now acceptable for publication, you may indicate that here to bypass the “Comments to the Author” section, enter your conflict of interest statement in the “Confidential to Editor” section, and submit your "Accept" recommendation.

Reviewer #3: All comments have been addressed

Reviewer #4: All comments have been addressed

Reviewer #5: All comments have been addressed

2. Is the manuscript technically sound, and do the data support the conclusions?

Reviewer #3: Yes

Reviewer #4: Yes

Reviewer #5: No

3. Has the statistical analysis been performed appropriately and rigorously? 

Reviewer #3: I Don't Know

Reviewer #4: Yes

Reviewer #5: Yes

4. Have the authors made all data underlying the findings in their manuscript fully available?

Reviewer #3: No

Reviewer #4: No

Reviewer #5: Yes

5. Is the manuscript presented in an intelligible fashion and written in standard English?

Reviewer #3: Yes

Reviewer #4: No

Reviewer #5: No

6. Review Comments to the Author

Reviewer #3: Overall, the authors addressed most of the suggestions. Although this is a small study, I commend the authors for evaluating this important topic of readmission after COVID-19 infection and its association with mortality.

I recommend that the authors add their explanation of why the risk of death increased with a history of cancer and hyperlipidemia. The discussion concentrated more on the cancer history and failed to elaborate on hyperlipidemia as a risk factor.

Reviewer #4: (No Response)

Reviewer #5: It can almost be said that the corrections made are good, but

The text of the article has not been written scientific

It is better to revise and edit

7. PLOS authors have the option to publish the peer review history of their article (what does this mean?). If published, this will include your full peer review and any attached files.

Reviewer #3: No

Reviewer #4: No

Reviewer #5: No

---

## [Author Response · Author response to Decision Letter 3]

25 Dec 2023

Comments to the authors: 

Overall, the authors addressed most of the suggestions. Although this is a small study, I commend the authors for evaluating this important topic of readmission after COVID-19 infection and its association with mortality. 

I recommend that the authors add their explanation of why the risk of death increased with a history of cancer and hyperlipidemia. The discussion concentrated more on the cancer history and failed to elaborate on hyperlipidemia as a risk factor.

Response: we tried to add some explanation about hyperlipidemia in the discussion as follows:

Furthermore, we found that hyperlipedemia increased the risk of mortality. It is concordant with other studies (25-27). Its possible mechanism is that when the lung is involved and the blood oxygen level decreases, one of the main compensatory mechanisms of the body to deliver enough oxygen to the tissues is to increase the cardiac output by increasing the heart rates and increasing the stroke volume. Naturally, endothelial dysfunction, atherosclerosis, and possible narrowing of coronary arteries in patients with hyperlipidemia, as well as probable more coronary artery diseases in them, can reduce the ability of the cardiovascular system to compensate tissue hypoxia in these patients. Also, three mechanisms of increased oxidative stress, pro-inflammatory conditions, and also risk of arrhythmia caused by hyperlipidemia (28) combined with COVID-19 can lead to increased mortality.

---

## [Decision Letter · Decision Letter 4]

6 Feb 2024

PONE-D-23-04746R4Comorbidities affecting re-admission and survival in COVID-19: Application of joint frailty modelPLOS ONE

Dear Dr. Shirafkan,

Thank you for submitting your manuscript to PLOS ONE. After careful consideration, we feel that it has merit but does not fully meet PLOS ONE’s publication criteria as it currently stands. Therefore, we invite you to submit a revised version of the manuscript that addresses the points raised during the review process.

We look forward to receiving your revised manuscript.

Kind regards,

Yee Gary Ang, MBBS MPH

Academic Editor

PLOS ONE

Journal Requirements:

**Additional Editor Comments:**

Please make the suggested changes. 

Reviewers' comments:

Reviewer's Responses to Questions

**Comments to the Author**

1. If the authors have adequately addressed your comments raised in a previous round of review and you feel that this manuscript is now acceptable for publication, you may indicate that here to bypass the “Comments to the Author” section, enter your conflict of interest statement in the “Confidential to Editor” section, and submit your "Accept" recommendation.

Reviewer #3: All comments have been addressed

Reviewer #6: (No Response)

2. Is the manuscript technically sound, and do the data support the conclusions?

Reviewer #3: Yes

Reviewer #6: Yes

3. Has the statistical analysis been performed appropriately and rigorously? 

Reviewer #3: I Don't Know

Reviewer #6: Yes

4. Have the authors made all data underlying the findings in their manuscript fully available?

Reviewer #3: Yes

Reviewer #6: (No Response)

5. Is the manuscript presented in an intelligible fashion and written in standard English?

Reviewer #3: Yes

Reviewer #6: Yes

6. Review Comments to the Author

Reviewer #3: The authors have addressed most recommendations and improved the manuscript. I commenced their persistence in analyzing and commitment to publishing the data.

Reviewer #6: Thank you for the opportunity to review this manuscript. Many reviews have been done on this article, which has made it better than its initial version.

this is a good study on the joint frailty model.

Some minor points to be considered and fixed before publication:

What do you mean confirming of covid-19 by imaging or laboratory documents In line of 116? CT scan and what laboratory finding ?

Why is NEWS2 finding not in the table? explanation in Line 190-193

It seems that the number of survivors In table 1 is wrong and should be corrected 64.

7. PLOS authors have the option to publish the peer review history of their article (what does this mean?). If published, this will include your full peer review and any attached files.

Reviewer #3: **Yes: **Monica I. Lupei

Reviewer #6: No

---

## [Author Response · Author response to Decision Letter 4]

26 Feb 2024

What do you mean confirming of covid-19 by imaging or laboratory documents In line of 116? CT scan and what laboratory finding?

Response:

Confirmation of COVID-19 was done by RT-PCR test (a laboratory document) or by imaging or laboratory documents like CRP)

Why is NEWS2 finding not in the table? explanation in Line 190-193

Response:

Should all the findings be mentioned again in the table? Considering that the severity of the disease was different in each hospitalization, we only evaluated the severity of the disease in the first visit and reported it in the text of the article.

It seems that the number of survivors In table 1 is wrong and should be corrected 64

Response:

We corrected it.

---

## [Decision Letter · Decision Letter 5]

12 Mar 2024

Comorbidities affecting re-admission and survival in COVID-19: Application of joint frailty model

PONE-D-23-04746R5

Dear Dr. Shirafkan,

We’re pleased to inform you that your manuscript has been judged scientifically suitable for publication and will be formally accepted for publication once it meets all outstanding technical requirements.

Kind regards,

Yee Gary Ang, MBBS MPH

Academic Editor

PLOS ONE

Additional Editor Comments (optional):

Reviewers' comments:

Reviewer's Responses to Questions

**Comments to the Author**

1. If the authors have adequately addressed your comments raised in a previous round of review and you feel that this manuscript is now acceptable for publication, you may indicate that here to bypass the “Comments to the Author” section, enter your conflict of interest statement in the “Confidential to Editor” section, and submit your "Accept" recommendation.

Reviewer #6: All comments have been addressed

2. Is the manuscript technically sound, and do the data support the conclusions?

Reviewer #6: Yes

3. Has the statistical analysis been performed appropriately and rigorously? 

Reviewer #6: Yes

4. Have the authors made all data underlying the findings in their manuscript fully available?

Reviewer #6: Yes

5. Is the manuscript presented in an intelligible fashion and written in standard English?

Reviewer #6: Yes

6. Review Comments to the Author

Reviewer #6: (No Response)

7. PLOS authors have the option to publish the peer review history of their article (what does this mean?). If published, this will include your full peer review and any attached files.

Reviewer #6: No

---

## [Editor Report · Acceptance letter]

21 Mar 2024

PONE-D-23-04746R5 

PLOS ONE

Dear Dr. Shirafkan, 

I'm pleased to inform you that your manuscript has been deemed suitable for publication in PLOS ONE. Congratulations! Your manuscript is now being handed over to our production team.

Kind regards, 

on behalf of

Dr. Yee Gary Ang 

Academic Editor

PLOS ONE